

# An experimental seasonal hydrological forecasting system over the Yellow River basin-Part II: The added value from climate forecast models

Xing Yuan[1]

[1]RCE-TEA, Institute of Atmospheric Physics, Chinese Academy of Sciences, Beijing, 100029, China

*Correspondence to*: Xing Yuan (yuanxing@tea.ac.cn)

**Abstract.** This is the second paper of a two part series on introducing an experimental seasonal hydrological forecasting system over the Yellow River basin in northern China. While the natural hydrological predictability in terms of initial hydrological conditions (ICs) is investigated in a companion paper, the added value from eight North American Multimodel Ensemble (NMME) climate forecast models with a grand ensemble of 99 members is assessed in this paper, with an implicit consideration of human-induced uncertainty in the hydrological models through a post-processing procedure. The forecast skill in terms of Anomaly Correlation (AC) for 2-m air temperature and precipitation does not necessarily decrease over leads, but is dependent on the target month due to a strong seasonality for the climate over the Yellow River basin. As there is more diversity in the model performance for the temperature forecasts than the precipitation forecasts, the grand NMME ensemble mean forecast has consistently higher skill than the best single model out to six months for the temperature, but up to two months for the precipitation. The NMME climate predictions are downscaled to drive the Variable Infiltration Capacity (VIC) land surface hydrological model and a global routing model regionalized over the Yellow River basin to produce forecasts of soil moisture, runoff and streamflow. And the NMME/VIC forecasts are compared with the Ensemble Streamflow Prediction method (ESP/VIC) through 6-month hindcast experiments for each calendar month during 1982-2010. As verified by the VIC offline simulations, the NMME/VIC is comparable to the ESP/VIC for the soil moisture forecasts, and the former has higher skill than the latter only for the forecasts at long leads and for those initialized in the rainy season. The forecast skill for runoff is lower for both forecast approaches, but the added value from NMME/VIC is more obvious, with an increase of the average AC by 0.08-0.2. To compare with the observed streamflow, both the hindcasts from NMME/VIC and ESP/VIC are post-processed through a linear regression model fitted by using VIC offline simulated streamflow. The post-processed NMME/VIC reduces the root mean squared error (RMSE) from the post-processed ESP/VIC by 5-15%. And the reduction occurs mostly during the transition from wet to dry seasons. With the consideration of the uncertainty in the hydrological models, the added value from climate forecast models is decreased especially at short leads, suggesting the necessity of improving the large-scale hydrological models in human intervened river basins.

1. **Introduction**



Seasonal climate forecasts have now been used to provide early warnings for health and food security (Thomson et al., 2006; Lizumi et al., 2013), and can be skilful for the applications that are mainly affected by temperature. However, due to a more chaotic nature and limited physical understanding, seasonal forecasting of precipitation has only marginal improvement (Smith et al., 2012; Saha et al., 2014), and the skill over land is not so favourable unless during the period with strong

oceanic anomalies like the El Niño (Stockdale et al., 1998). An intermediate solution is the ensemble forecasting technique, including the ensembles of different initial conditions by perturbing Sea Surface Temperature (SST) and wind stress (Slingo and Palmer, 2011) or by running the climate model with different start dates (Saha et al., 2014), as well as the ensembles from multiple climate forecast models (Krishnamurti et al., 1999). Ensembles of initial conditions based on a single model do not necessarily sample the forecast space completely, and usually result in under-dispersion errors. Therefore, multimodel

ensemble forecasts are receiving more attentions from a variety of perspectives, including the applications in the hydrological forecasting (Luo and Wood, 2008; Pappenberger et al., 2008; Demargne et al., 2014; Yuan et al., 2015a).

In fact, multimodel ensemble weather forecasts have already been successfully used for short-term hydrological forecasts. For example, Pappenberger et al. (2008) found that if the grand THORPEX International Grand Global Ensemble (TIGGE) forecasts had been used, flood warnings could be issued 8 days before the event, whereas the warning based on a single

ensemble system would only allow for a lead time of 4 days. The continuation of the TIGGE project (Swinbank et al., 2016) will further benefit the flooding forecasts. Similarly, the seasonal climate prediction from the Development of a European Multimodel Ensemble System for Seasonal-to-Interannual Prediction (DEMETER) project (Palmer et al., 2004) was used to improve the hydrological forecasting over the Ohio River basin during the first two months (Luo and Wood, 2008). However, as compared with the short-term flood forecasting (Pappenberger et al., 2005; Cloke and Pappenberger, 2009), the seasonal

hydrological forecasting based on multiple climate forecast models is less widely applied in general. One of the reasons is that it is difficult to find the added value from climate-model-based seasonal hydrological forecasting as compared with the traditional Ensemble Streamflow Prediction (Day, 1985) method because significant climate prediction skill that is useful for hydrological forecasting is usually regime-dependent (Wood et al., 2002; Luo and Wood, 2007; Mo et al., 2012; Sinha and Sankarasubramanian, 2013; Yuan et al., 2013; Shukla et al., 2014; Trambauer et al., 2015). Another important reason is the

lack of an open source of multimodel seasonal climate hindcast datasets that can be used to understand the hydro-climate predictability from global to river basin scales and to develop climate-model-based experimental or operational seasonal hydrological forecasting systems for an adaptive hydrological service.

Since 2011, the National Oceanic and Atmospheric Administration (NOAA)'s Modeling, Analysis, Prediction, and Projections (MAPP) program has been supporting the implementation and assessment of an experimental North American

Multimodel Ensemble (NMME; Kirtman et al., 2014) seasonal forecast system as part of the NOAA Climate Test Bed and Climate Prediction Task Force research (Wood et al., 2015). Several decades of NMME hindcast datasets are available for the public research community, which provides an unprecedented opportunity to assess the added value for seasonal hydrological forecasting. In addition, the NMME is now being made to produce global seasonal climate prediction in a real-



time mode, which motivates the development of experimental seasonal hydrological forecasting systems based on the downscaled NMME prediction at regional, continental and global scales.

Recently, a few studies have been carried out to investigate the usefulness of the NMME in advancing seasonal hydrological forecasting. Driving a hydrological model with the NMME seasonal climate hindcasts, Mo and Lettenmaier (2014) analyzed the skill of monthly and seasonal soil moisture and runoff forecasts over the United States by comparing with the ESP-based forecasts, and found that the climate forecasts contribute to the hydrological forecast skill over wet regimes. Thober et al. (2015) used similar method to assess the soil moisture drought prediction over the Europe, and found that the NMME-based method outperforms the ESP-based method for drought forecasting at all lead times. Besides continental-scale hydrological forecasting, Yuan et al. (2015a) assessed the value of NMME in improving the seasonal forecasting of hydrological extremes over global major river basins, and the NMME/hydrology method showed higher detectability for soil moisture drought, more reliable low and high flow ensemble forecasts as compared with the ESP approach.

However, even the state-of-the-art NMME climate predictions could not help the hydrological forecasting over the river basins with limited hydrological gauges and less reliable meteorological observations that are used to correct the errors in the hydrological model and climate prediction (Sikder et al., 2016). In addition, most NMME/hydrology assessments neglected the uncertainty in hydrological model for the forecast verification (Mo and Lettenmaier, 2014; Thober et al., 2015; Yuan et al., 2015a; Sikder et al., 2016), except for an assessment for a "real-time" forecasting of the 2012 North American drought where the model-predicted soil moisture drought area is verified against the satellite retrievals (Yuan et al., 2015a). As shown by Yuan et al. (2013), the added value from climate-model-based streamflow forecasting tends to diminish over some river basins if the observed streamflow instead of the simulate streamflow is used for forecast verification. For those river basins, uncertainty in the hydrological modeling might be larger than the uncertainty in climate forecast at short leads, or the error in the hydrological model might be too large to reflect the improved skill in precipitation. Actually, Yuan and Wood (2012) discussed whether the downscaling of climate prediction or the bias-correction of streamflow is more important for the seasonal streamflow forecasting, and they hypothesized that the errors in the climate prediction could be amplified through the nonlinear rainfall-runoff processes and resulted in a unreliable streamflow forecast even if the climate prediction had been corrected as reliable. Therefore, a hydrological post-processor that is used to correct the errors in hydrological models and/or the propagation of climate forecast errors is essential to seasonal hydrological forecasting, especially over those river basins with heavy human interventions.

The Yellow River basin is a heavily managed river basin located in northern China. The surface water resources in the Yellow River account for only about 2% of total surface water resources in China, but they are used to irrigate 15% of the cropland and to raise 12% of the population in China. Before establishing an operational forecasting system that can handle the detailed physical processes of irrigation and inter-basin water diversion in a climate-hydrology coupled mode that is currently not available due to the scarcity of management data and the deficiency in the human component in most hydrological models, it is necessary to understand the naturalized hydrological predictability and the added value from climate forecast models by using an experimental seasonal hydrological forecasting system over the Yellow River basin, and



to use the hydrological post-processing as an intermediate approach to account for the human interventions implicitly in the forecasting system.

The first paper of the two-part series introduced the climate-hydrology forecasting system and investigated the naturalized hydrological predictability in terms of initial hydrological conditions through the reverse ESP-type simulation (Yuan et al.,

2016). This paper focuses on the evaluation of the NMME-based seasonal hydrological forecasting by comparing with the ESP approach over the Yellow River basin. Besides assessing the added value from climate forecast models by neglecting the errors in the hydrological models (i.e., verifying the hydrological forecasts with model offline simulations driven by observed meteorological forcings), this paper also tries to evaluate the seasonal forecast skill in a "real" world by using a hydrological post-processing procedure.

2. **Data and Method**

**2.1 Downscaling of NMME climate prediction**

As described in the companion paper (Yuan et al., 2016), hydrometeorological datasets from 324 meteorological stations and 12 mainstream hydrological gauges are used to calibrate the Variable Infiltration Capacity (VIC; Liang et al. 1996) land surface hydrological model and a global routing model (Yuan et al., 2015a) regionalized over the Yellow River. To our

understanding, this is the first time that over three hundred meteorological station observations have been used to study the hydrological forecasting over the Yellow River. The improved quality of the meteorological observations not only facilitates a more objective calibration of the hydrological models, but also helps the downscaling and bias correction of the seasonal climate predictions. The meteorological datasets for precipitation, 2-m maximum and minimum air temperature and 10-m wind speed are interpolated into 1321 grid cells at a 0.25-degree resolution, with a lapse rate correction for temperature at

different elevations.

In this study, eight NMME models with 99 realizations in total (Table 1) are used for the seasonal hydrological forecasting. The NMME leverages considerable research and development activities on coupled model prediction systems carried out at universities and various research laboratories throughout North America (Kirtman et al., 2014). Besides using the NMME hindcasts for hydrological forecasting over the USA, Europe, south Asia and global major river basins (Mo and Lettenmaier,

2014; Thober et al., 2015; Yuan et al., 2015a; Sikder et al., 2016), the NMME was also used to assess the potential drought predictability over China (Ma et al., 2015). Given that one of the NMME models, the NCEP-CFSv2, has an ensemble with different initialization dates (Saha et al., 2014), the month-1 forecast is called as a forecast at 0.5 month lead, and the month-2 is at 1.5 month lead, and so on.

Similar to Yuan et al. (2015a), the NMME hindcasts are downscaled and bias-corrected through the quantile-mapping

method (Wood et al., 2002) as follows: 1) the 1-degree NMME global hindcasts of monthly precipitation and temperature during 1982-2010 are first bilinearly interpolated into 0.25-degree over the Yellow River; 2) for each calendar month and each NMME model, all hindcasts (excluding the target year) with all ensemble members for the target month are used to construct cumulative distribution functions (CDFs) of the forecasts, the CDFs of observations are constructed similarly (excluding the target year), and the hindcast in the target year is adjusted by matching its rank in the CDF of the forecasts





and that in the CDF of the observations to remove the bias; and 3) the bias-corrected monthly hindcasts of precipitation and temperature are temporally downscaled to a daily time step by sampling from the observation dataset and rescaling to match the monthly hindcasts.

## 2.2 Hydrological post-processing

The downscaled NMME climate predictions are used to drive the VIC land surface hydrological model to provide soil moisture and runoff forecasts up to six months, and the runoff forecasts are used to drive the routing model to provide streamflow forecasts. The results represent "naturalized hydrological forecasts" because the hydrological models were calibrated against naturalized streamflow as described in Yuan et al. (2016). To make the forecasts comparable to the hydrological observations over the Yellow River where human interventions occur at middle and lower reaches, a hydrological post-processing procedure is necessary to correct the raw forecasts without human components. In this study, a linear regression is applied to correct the streamflow forecasts at 12 mainstream gauges where the observations are available. The regression coefficients are firstly fitted between observed and offline simulated streamflow for each calendar month, then the coefficients are applied to correct the streamflow forecasts for their target months.

Table 2 lists the Nash-Sutcliffe efficiency (NSE) for the post-processed streamflow simulations during 1982-2010. As compared with the results that are verified by using the naturalized streamflow, the NSE values decrease by 0.1-0.4 (except for the Tangnaihai gauge at the headwaters region where almost no human interventions occur). However, there are many negative NSE values without implementing the post-processing procedure (not shown), which is because of large systematic biases in the simulations neglecting the processes such as irrigation water withdraw. Therefore, the post-processing is an effective intermediate method to reduce the uncertainty in the hydrological modelling. In fact, the NSE averaged among the 12 gauges is about 0.61, which is still much higher than the climatology (with NSE=0). For the Tangnaihai gauge in the headwaters region, the naturalized streamflow is almost the same as the observed streamflow, so a higher NSE after post-processing (Table 2) indicates that the post-processing can also reduce the errors in hydrological modelling that is less relevant to human intervention. In other words, the post-processing procedure reduces both the "natural" and "anthropogenic" errors in the hydrological model in an integral manner.

## 2.3 Experimental design and evaluation metrics

As described in Yuan et al. (2016), a continuous offline hydrological simulation driven by observed meteorological forcings from 1951 to 2010 was conducted to generate the initial hydrological conditions (ICs) for the VIC land surface hydrological model and the river routing model, and the 6-month ESP/VIC hydrological hindcasts with 28 ensemble members during 1982-2010 were carried out to provide a reference forecast. The NMME/VIC hindcasts use the same ICs as the ESP/VIC, i.e., those generated by the offline simulations, and use meteorological hindcasts from eight NMME models. The grand NMME/VIC ensemble is an average of 99 ensemble hydrological hindcasts.

One of the measures for assessing the hydroclimate forecast skill is the anomaly correlation (AC; Wilks, 2011), which is defined as:





$$AC = \frac{\sum\sum X'(s,t)Y'(s,t)}{\left[\sum\sum X'(s,t)^2 \bullet \sum\sum Y'(s,t)^2\right]^{1/2}}, \tag{1}$$

where $X'(s,t)$ is the hydrological forecast, and $Y'(s,t)$ is the verification data; for a given lead and forecast target month/season, the summation is both over time ($t$, 29 years in this study) and space ($s$, 1321 grid cells for the Yellow River basin). If the AC is used for each grid cell within the Yellow River basin (i.e., there is only a summation over time), it is

reduced to the Pearson correlation.

Another measure to determine whether the target forecast (NMME/VIC) is more skilful than the reference forecast (ESP/VIC) is the root mean squared error skill score ($SS_{RMSE}$; Wilks, 2011). The $SS_{RMSE}$ is defined as 1-$RMSE_{NMME}/RMSE_{ESP}$, where $RMSE_{NMME}$ and $RMSE_{ESP}$ are the root mean squared errors for NMME/VIC and ESP/VIC forecasts respectively. Here, $SS_{RMSE}$=1 indicates a perfect forecast, while $SS_{RMSE}$ less than zero means that the NMME/VIC

forecast is worse than ESP. Unless otherwise specified, the ensemble mean for ESP, individual climate models and the grand NMME mean are used for the skill assessment.

### 3.   Temperature and Precipitation Forecast Skill

Figure 1 shows the skill of monthly mean (ensemble mean) surface air temperature at 2-m above ground over the Yellow River basin. The X axis is the target or verification month, and the Y axis is the forecast lead in months. For example,

forecasts for June at a lead of 3.5 month for the COLA-RSMAS-CCSM4 have an AC around 0.35 (Fig. 1a), they are for the forecasts initialized in March but verified at June. Most climate models show a forecast skill that is not necessarily lower at longer leads, but is dependent on the target month. For example, Figure 1b shows that the GFDL-CM2p1 model has a low skill in the first month (less than 0.2) for the forecasts initialized in May, but the skill increases to 0.35 in the second month (June). Similar skill dependence on the target month can be found for another two GFDL models (Figs. 1c-1d) for March and

June, and the NCEP-CFSv2 model for the summer time (Fig. 1f) etc. For the GFDL models at higher resolution (Figs. 1c-1d), the skill is low during the first month, but the skill increase at longer leads. This might be caused by the initialization procedure over land because these two models are experimental forecasting models.

For the forecasts in the first month, the NCEP-CFSv2 has the highest skill in general (Fig. 1f), with an average AC of 0.46. However, other models have the best forecast skill for a specific month/season. For instance, the COLA-RSMAS-CCSM4

has higher forecast skill than the NCEP-CFSv2 for the forecasts of November at 0.5 month lead (Fig. 1a). Such complementary feature is more obvious for the forecasts at long leads. As a result, the grand NMME ensemble mean forecast (Fig. 1i) has consistently higher skill than the best single model, with an average AC of 0.5 at 0.5 month lead, and about 0.3 up to 6 months. Figure 1i shows that the highest forecast skill for 2-m temperature occurs during the summer and late winter, and the lowest skill occurs during the late spring. The low temperature forecast skill for the spring months at long leads

might be related to the snow processes during the early winter.

Figure 2 shows similar plots for the precipitation forecasts. Again, the forecast skill does not necessarily decline over leads due to a strong seasonality in the precipitation. The NASA-GMAO is the best model for the precipitation forecast at 0.5



month lead (Fig. 2e), with an average AC of 0.31. The NCEP-CFSv2 starts to rank the first for the forecast at 1.5 month and beyond (Fig. 2f), with average ACs of 0.06-0.08. The grand NMME ensemble for precipitation forecast (Fig. 2i) has a higher skill than individual models during the first 2 months, with average ACs of 0.35 and 0.09 at 0.5 and 1.5 month leads respectively. Beyond the first two months, the forecast skill of NMME is comparable to the best single model (i.e., NCEP-

CFSv2), but both have an AC lower than 0.1. One may wonder about the significance of the low correlations. The uncertainty (sampling error) in a correlation is $1/\sqrt{N-2}$, where $N$ is the effective number of cases. For the AC over the Yellow River basin, the $N$ is 29 (years) × 1321 (grid cells) = 38309, so an AC of 0.05 would be enough for the statistical significance. However, this does not mean that the low correlation is practically useful.

Figure 3 shows the spatial distribution of AC for the grand NMME ensemble forecasts for the precipitation averaged over

the first season. As described in section 2.3, the grid-scale AC reduces to the Pearson correlation. And given that the hindcast period is 1982-2010, the correlation is significant if it is larger than 0.37 (0.31) at the 5% (10%) level. For the upper reaches of the Yellow River, there is significant forecast skill at the beginning of the cold season (Figs. 3i-3j). For the middle and lower reaches, forecasts starting from November have the highest skill (Fig. 3k). During the spring, the forecasts are skilful over the northern part (Fig. 3c-3e). And during the summer, the forecasts are skilful over a marginal wet region in the

southern part of the Yellow River, with correlations higher than 0.37 (p<0.05).

## 4.   Soil Moisture and Streamflow Forecast Skill

The precipitation and temperature forecasts with 99 NMME ensemble members are downscaled and used to drive the VIC land surface hydrological model to provide seasonal hydrological forecasts. The grand NMME/VIC ensemble mean values are used for the analysis hereafter. Figure 4 shows the AC of soil moisture and runoff ensemble mean forecasts from

ESP/VIC and NMME/VIC, where the forecasts are verified against offline simulations. Unlike the precipitation and temperature forecasts that the skill does not necessarily decline over leads, the forecast skill for soil moisture and runoff generally decreases as the forecast leads proceed, especially during the dry seasons. This indicates that the ICs have strong impacts on the forecast skill for the land surface variables. The skill is very high for the soil moisture forecasts, especially for the target months during winter and spring (Figs. 4a-4b). The AC averaged over 12 target months for the ESP/VIC soil

moisture forecasts is higher than 0.8 out to three months. The NMME/VIC shows no improvement against the ESP/VIC in cold seasons given the strong memory of the soil moisture. However, the added value occurs for the target months in autumn at long leads, i.e., NMME/VIC can improve the skill for the forecasts initialized in the rainy season, and the improvement becomes more obvious after the rainy season (Figs. 4a-4b).

Figure 4c shows that ESP/VIC has lower forecast skill for the runoff than that for the soil moisture. The AC averaged over

12 target months for the ESP/VIC runoff forecasts is 0.64 at 0.5 month lead, drops to 0.2 at 2.5 month lead, and even becomes negative after the first 4 months. As the ICs have less control on the runoff forecasts, the added value from climate forecast models becomes more obvious. The skill for the runoff forecasts from NMME/VIC is consistently higher than that from the ESP/VIC, especially for the target months from late spring to early autumn (Figs. 4c-4d). The AC averaged over 12





target months for the NMME/VIC runoff forecasts is 0.72 at 0.5 month lead, drops to 0.38 at 2.5 month lead, and keeps a value larger than 0.2 out to 6 months. Therefore, NMME/VIC increases the average AC by 0.08-0.2, and the increase is larger at long leads.

Figure 5 shows the spatial distributions of the correlations for the soil moisture forecasts. For each grid cell, the correlation is an average of 12 target months. Similar to the predictability analysis in Yuan et al. (2016), strong soil moisture memory exists over the middle reaches of the Yellow River, with an averaged correlation higher than 0.5 out to 6 months for the ESP/VIC soil moisture forecasts (Figs. 5a-5c). For the upper and part of the lower reaches, there are no significant correlations for the ESP/VIC forecasts beyond 3 months (Figs. 5b-5c). As a result, significant improvements from NMME/VIC for the soil moisture forecast mainly occur over the upper and lower reaches of the Yellow River at a lead beyond 2 months (Figs. 5d-5f).

Figure 6 shows similar average correlation plots, but for the streamflow along the mainstream and major tributaries of the Yellow River. Given that the ICs control the first month streamflow forecasting greatly, ESP/VIC has an average correlation that is higher than 0.7 for the streamflow forecasts along the mainstream at 0.5 month lead (Fig. 6a), and there is only a marginal improvement from the NMME/VIC at upper reaches of the mainstream and tributaries (Fig. 6d). Beyond the first month, the added value from the NMME/VIC emerges, with an average correlation consistently higher than the ESP/VIC along the mainstream and major tributaries (Figs. 6b-6c, 6e-6f). The NMME/VIC increases the correlation for the streamflow forecast by 0.1-0.4, and the increase is more significant at long leads.

## 5. The Impact of Hydrological Post-processing

The above section shows the evaluation against model offline simulations of soil moisture, runoff and streamflow, i.e., it explores the added value from climate forecast models by neglecting the errors in the hydrological models. To go one step further, it is necessary to assess the climate-model-based seasonal hydrological forecasting with the consideration of the uncertainty in the hydrological models. Therefore, the hydrological forecasts should be verified with the observations. In terms of runoff, there is no direct observation at a large scale. The runoff is usually derived from water balance models, or obtained from the inverse streamflow routing through the data assimilation method (Pan and Wood, 2013). But again these estimates are more or less a model product. The soil moisture can be measured at local scale, but again its representative at a large scale is questionable given the strong heterogeneity of the land surface. The satellite remote sensing is a promising method to measure the soil moisture at a large scale, but currently its quality on representing the short-term variability is still a concern (Yuan et al., 2015b). Different from runoff and soil moisture, the streamflow can be measured at a certain gauge for a certain drainage area. Therefore, the streamflow forecasts both from ESP/VIC and NMME/VIC are verified with observation after the post-processing procedure described in section 2.2.

Figure 7 shows the time series of the post-processed model streamflow and the observed streamflow at five hydrological gauges from the upper to lower mainstream of the Yellow River. As compared with the naturalized streamflow (Figure 4 in Yuan et al. (2016)), the observed streamflow shows a nonstationary feature, suggesting a human perturbation combined with the climate change impact over the Yellow River basin. After post-processing, the VIC simulated streamflow matches with



the observation quite well at the upper gauges, but has a weaker decadal change during the 1980s and 1990s for the lower gauges.

Figure 8 shows the RMSE skill score for the streamflow forecasts at 12 mainstream gauges, without considering the error in the hydrological models. The reference forecast is the ESP/VIC, and a skill score above zero represents the added value from climate forecast models. Figure 9 shows similar plots, but the RMSEs are calculated between post-processed forecasts and the observed streamflow. Regardless of the errors in the hydrological models, the NMME reduces the RMSE for the streamflow forecasts by 10-25% (Fig. 8). As compared with the observed streamflow, the NMME reduces the RMSE by less than 5-15% (Fig. 9). And the reduction occurs mostly during the transition from wet to dry seasons. The RMSE skill score generally decreases over leads without the consideration of the errors in hydrological models (Fig. 8), but it may increase as verified by the observed streamflow (Fig. 9). This suggests that the added value from climate models at a long forecast lead might not be negligible as we expected, or they might be underestimated by previous studies that verify the forecasts with model simulations.

## 6. Concluding Remarks

This is the second paper of a two-part series on introducing an experimental seasonal hydrological forecasting system over the Yellow River basin in northern China. The system downscales the seasonal climate forecasts from the North American Multimodel Ensemble (NMME) models, and drives the Variable Infiltration Capacity (VIC) land surface hydrological model and a global routing model regionalized over the Yellow River basin to produce seasonal hydrological forecasting of soil moisture, runoff and streamflow at a 0.25-degree resolution. The first paper investigates the hydrological predictability in terms of initial hydrological conditions (ICs) by performing the reverse Ensemble Streamflow Prediction (revESP) simulations using the hydrological models in the forecasting system. This paper evaluates the added value for the seasonal hydrological forecasting from climate forecast models by using 99 ensemble forecasts of surface air 2-m temperature and precipitation from eight NMME models during 1982-2010, as compared with ESP-type forecasts.

The forecast skill in terms of Anomaly Correlation (AC) for 2-m temperature and precipitation does not necessarily decrease over leads, but is dependent on the target month due to a strong seasonality for the climate over the Yellow River basin. The highest forecast skill for 2-m temperature occurs during the summer and late winter, and the lowest skill occurs during the late spring. Among eight NMME models used in this study, the NCEP-CFSv2 and NASA-GMAO models have the highest AC for the 2-m temperature and precipitation forecasts at the first month respectively. After the first month, the skill for NCEP-CFSv2 is consistently higher than other NMME models for the precipitation forecasts, but not for the temperature forecasts. As there is more diversity in the model performance for the temperature forecasts, the grand NMME ensemble mean forecast has consistently higher skill than the best single model, with an average AC of 0.5 for the 0.5 month lead, and about 0.3 up to 6 months. For the precipitation forecasts, the grand NMME ensemble mean forecast has higher skill than the best individual models during the first two month, and its skill is comparable to the best individual model beyond the first two months. During the first season, the NMME ensemble mean precipitation forecasts have statistically significant skill over northern part of the Yellow River basin for the forecasts initialized in spring, over southern marginal regions with wet



climate for the forecasts initialized in summer, over the upper reaches for the forecasts initialized at the beginning of the cold season, and over the middle and lower reaches for that initialized in November.

Due to the dominant role of ICs in the forecasting of land surface conditions, the forecast skill for soil moisture and runoff as verified with offline VIC simulation without considering the model errors, decreases generally as the lead increases

especially during the dry seasons. The soil moisture forecast skill for the ESP method is very high, with an averaged AC among 12 target months higher than 0.8 out to three months. The NMME climate models can improve the forecast skill against the ESP for the forecasts at long leads and for those initialized in the rainy season. As the ICs have weaker control on the runoff than the soil moisture, the added value from climate forecast models is more obvious for the runoff forecasts. Compared with the ESP/VIC runoff forecasts, the NMME/VIC increases the average AC by 0.08-0.2, and the increase is

larger at long leads. In terms of spatial distributions, both the ESP/VIC and NMME/VIC have high forecast skill for the soil moisture over the middle reaches. The later increases the average AC from the former by 0.08-0.2 over upper and lower reaches of the Yellow River basin, and the increase is larger at long leads. For the streamflow forecasting, the ESP/VIC has an averaged correlation higher than 0.7 along the mainstream at 0.5 month lead, where there is only a marginal improvement from NMME/VIC at upper reaches and tributaries. However, the NMME/VIC increases the correlation for the streamflow

forecasts at long leads by 0.1-0.4.

The NMME/VIC reduces the root mean squared error (RMSE) from ESP/VIC by 10-25% across all target months for the streamflow forecasts verified by neglecting the uncertainty in hydrological models (i.e., verified by the offline simulated streamflow). To compare with the observed streamflow, the predicted streamflow from both ESP/VIC and NMME/VIC are post-processed through a linear regression, with the regression model fitted by offline simulation results. As verified by

observed streamflow, the NMME/VIC reduces the RMSE from ESP/VIC by 5-15%, especially during the transition from wet to dry seasons. Regardless of the errors in hydrological models, the added value from climate forecast models decreases over leads, which is consistent with the increase of error in the climate forecast. However, with the consideration of the uncertainty in the hydrological models, the added value from climate model may increase over leads, which suggests that the usefulness of the climate forecasts in the hydrological forecasts at long leads might be underestimated in the studies that

verifies the forecasts with model offline simulations.

This study shows that the NMME-based forecasting outperforms the ICs-based forecast method over the Yellow River basin, with or without the consideration of the errors in the hydrological models. Toward establishing an operational seasonal hydrological forecasting system, future efforts could be spent as follows: (1) explicitly representing the human intervention processes in the forecasting system is not only necessary to further reduce the uncertainty in the hydrological models, but

also to facilitate the understanding of the hydrological predictability with human dimension; (2) for the variables that are not easily to be corrected due to limited observations (e.g., soil moisture, runoff), forecasting with multiple hydrological models might be useful to quantify the uncertainty; (3) there is a decadal variation for the observed streamflow over the Yellow River basin, which is a result of both decadal climate change and the human water use change such as the water allocation in the 1980s, and water conservation through planting more trees over the Loess Plateau. Attribution of the natural and



anthropogenic changes in the environment and assessing their impacts on the terrestrial hydrology are not only interesting questions within the scope of the global change, but are also relevant for developing the short-term hydrological forecasting systems because they will influence the downscaling statistics, the calibration of hydrological models, and the hydrological post-processing. Therefore, more collaborations between the climate research scientists and operational hydrological

forecasters should be put on the agenda, and the Global Framework for Climate Services (GFCS) is a good concept that facilitates the transfer of the advances in climate research to climate services including the seasonal hydrological forecasting that is targeted for adaption to hydrologic extremes; and (4) given that ensemble seasonal hydrological forecasting becomes popular, it is the time to think about the interpretation of the ensemble forecast results to the decision makers (Hoss and Fischbeck, 2016). A useful ensemble forecast should be reliable but also sharper than a climatological forecast (toward a

more deterministic forecast), which is not always the case. There should be a balance between the reliability and the sharpness, and how to determine an effective balance is a question both for scientists and managers.

**Acknowledgement.** This work was supported by the National Natural Science Foundation of China (No. 91547103), and the Thousand Talents Program for Distinguished Young Scholars. We would like to acknowledge the NMME project and the

data dissemination supported by NOAA, NSF, NASA and DOE with the help of IRI personnel.





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





**Table 1.** List of NMME models used in this study.

| Model | Version | Ensemble |
|---|---|---|
| Community Climate System Model | 4 (COLA-RSMAS-CCSM4) | 10 |
| Geophysical Fluid Dynamics Laboratory Climate Model | 2.1 (GFDL-CM2p1) | 10 |
| | 2.5 (GFDL-CM2p5-FLOR-A06) | 12 |
| | 2.5 (GFDL-CM2p5-FLOR-B01) | 12 |
| Goddard Earth Observing System Model | 5 (NASA-GMAO) | 11 |
| Climate Forecast System | 2 (NCEP-CFSv2) | 24 |
| Canadian Coupled Global Climate Model | 3 (CMC1-CanCM3) | 10 |
| | 4 (CMC2-CanCM4) | 10 |

**Table 2.** Information at twelve hydrological gauges and the Nash-Sutcliffe efficiency (NSE) verified by using the naturalized and observed streamflow during 1982-2010. When it verified against the observed streamflow, the simulated streamflow is post-processed before calculating the NSE.

| Gauge | Latitude (°N) | Longitude (°E) | Drainage Area ($10^3$ km$^2$) | NSE with naturalized streamflow | NSE with observed streamflow |
|---|---|---|---|---|---|
| Tangnaihai | 35.5 | 100.15 | 122 | 0.87 | 0.91 |
| Xunhua | 35.83 | 102.5 | 145 | 0.88 | 0.42 |
| Xiaochuan | 35.93 | 103.03 | 182 | 0.84 | 0.58 |
| Lanzhou | 36.07 | 103.82 | 223 | 0.91 | 0.67 |
| Xiaheyan | 37.45 | 105.05 | 254 | 0.90 | 0.63 |
| Shizuishan | 39.25 | 106.78 | 309 | 0.89 | 0.58 |
| Hekouzhen | 40.25 | 111.17 | 428 | 0.76 | 0.53 |
| Longmen | 35.67 | 110.58 | 498 | 0.74 | 0.55 |
| Sanmenxia | 34.82 | 111.37 | 688 | 0.77 | 0.63 |
| Huayuankou | 34.92 | 113.65 | 730 | 0.81 | 0.57 |
| Gaocun | 35.38 | 115.08 | 734 | 0.78 | 0.59 |
| Lijin | 37.52 | 118.3 | 752 | 0.71 | 0.63 |





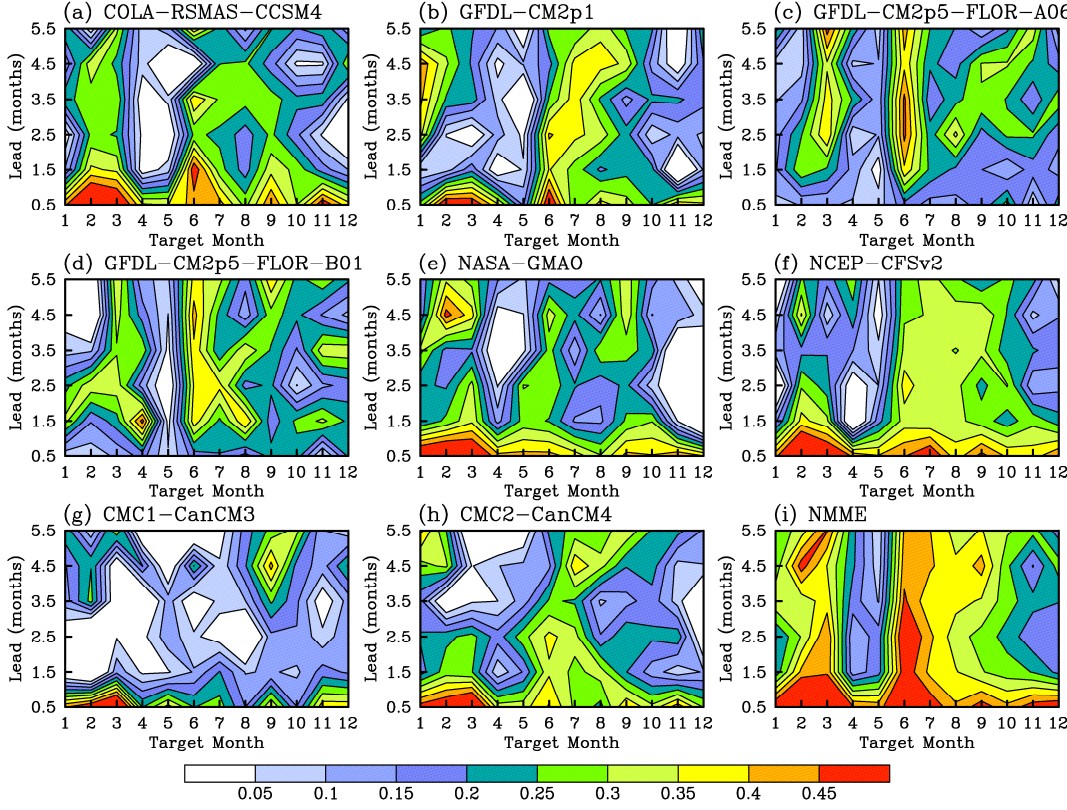

**Figure 1.** Anomaly Correlation (AC) of ensemble mean forecasts from eight NMME models (a-h) and the grand NMME ensemble averaged among 99 realizations as a function of lead and target month for monthly mean 2-m temperature over the Yellow River basin during the period of 1982-2010.




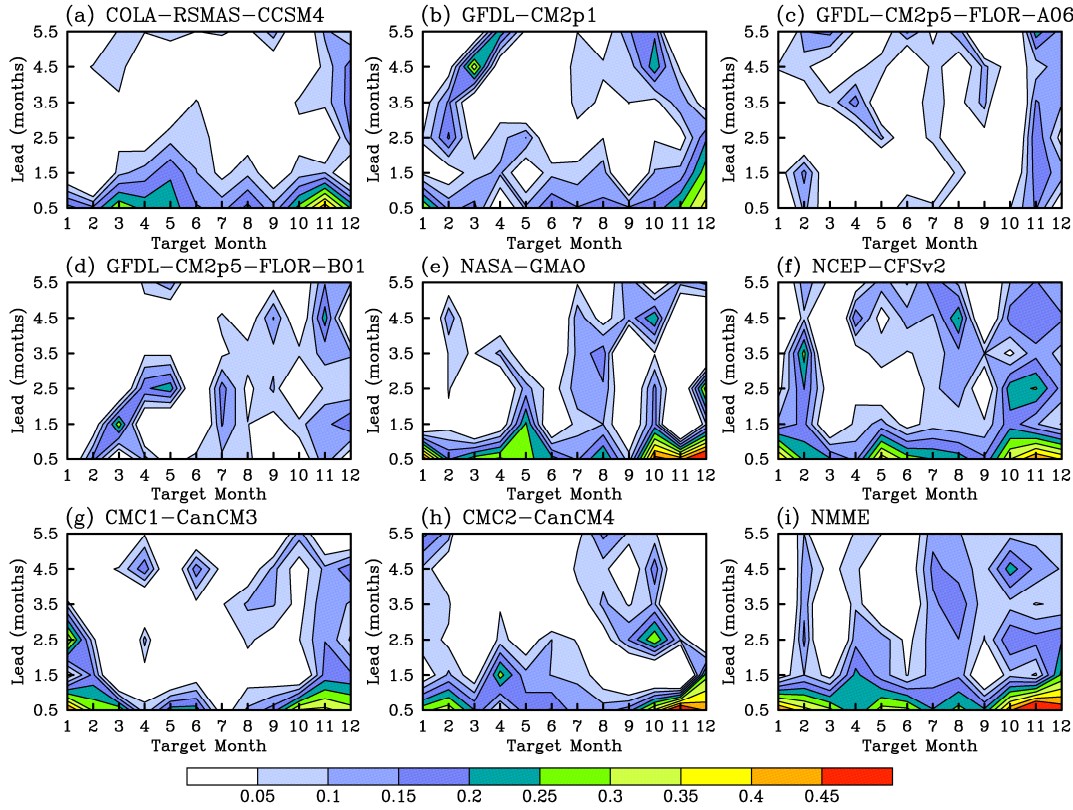

**Figure 2.** The same as Figure 1, but for monthly mean precipitation.





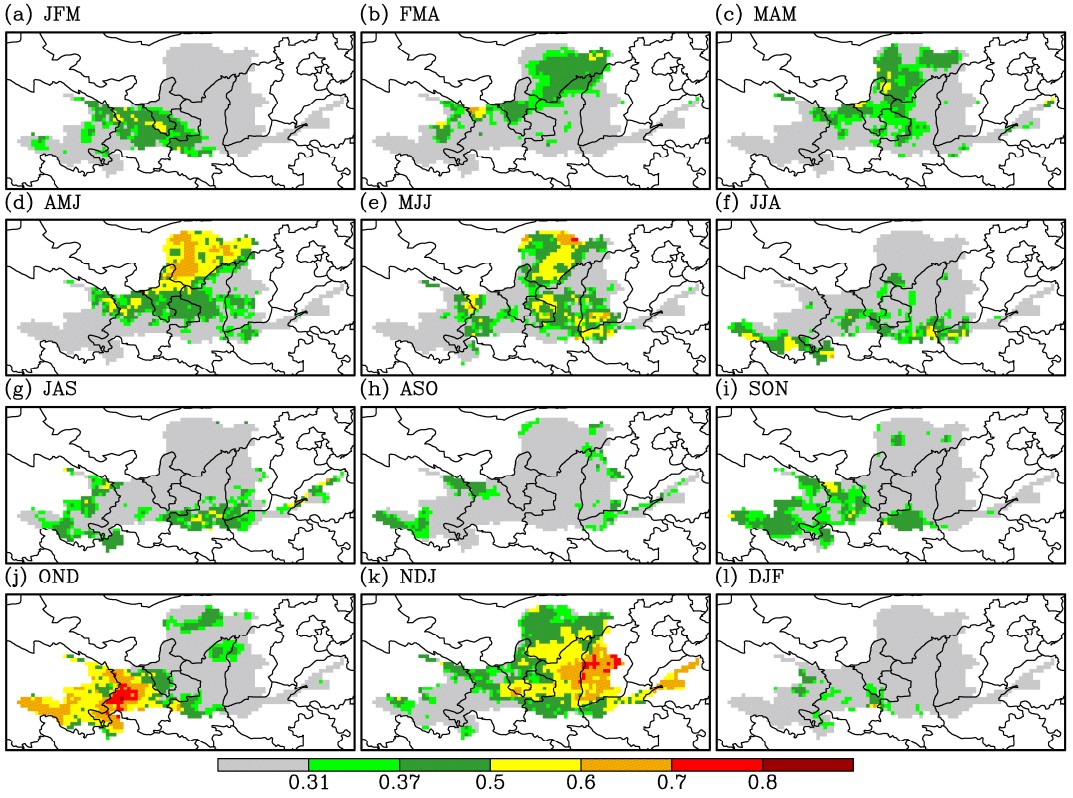

**Figure 3.** Spatial distributions of grid-scale AC of grand NMME ensemble forecasts for seasonal mean precipitation.





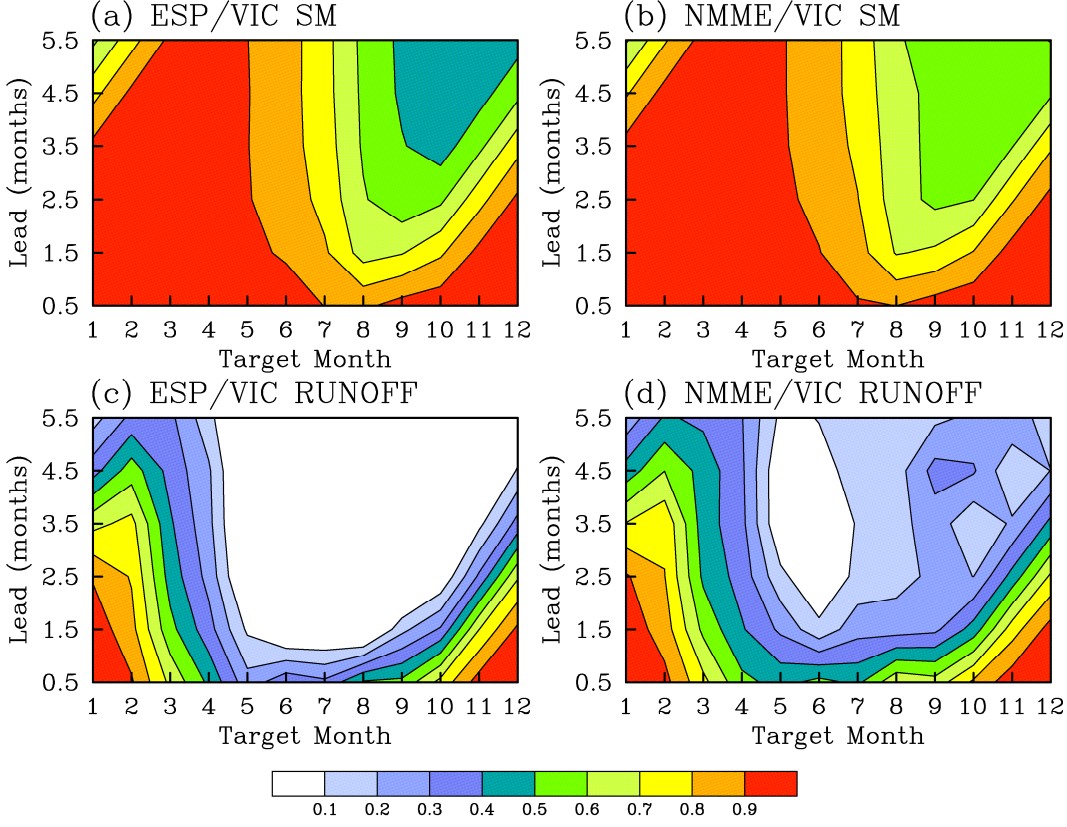

**Figure 4.** AC of ensemble mean hydrological forecasts from a climatology method (ESP/VIC) and the climate-model-based approach (NMME/VIC) as a function of lead and target month for monthly mean soil moisture (a-b) and runoff (c-d) over the Yellow River basin during the period of 1982-2010. The soil moisture and runoff used for the verification are from VIC offline simulation.





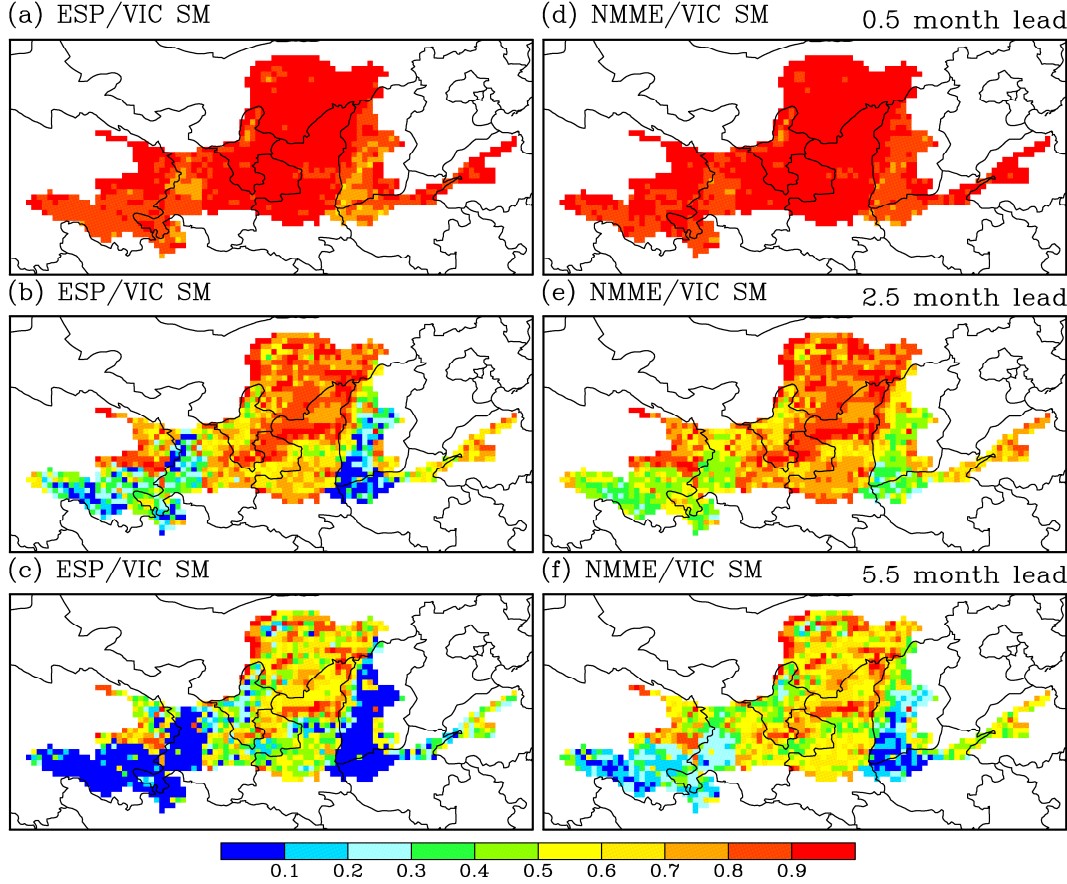

**Figure 5.** Spatial distributions of average AC of ensemble mean forecasts from ESP/VIC (left panel) and NMME/VIC (right panel) for monthly soil moisture at different leads. The average AC is the mean for the forecasts starting from twelve target months.





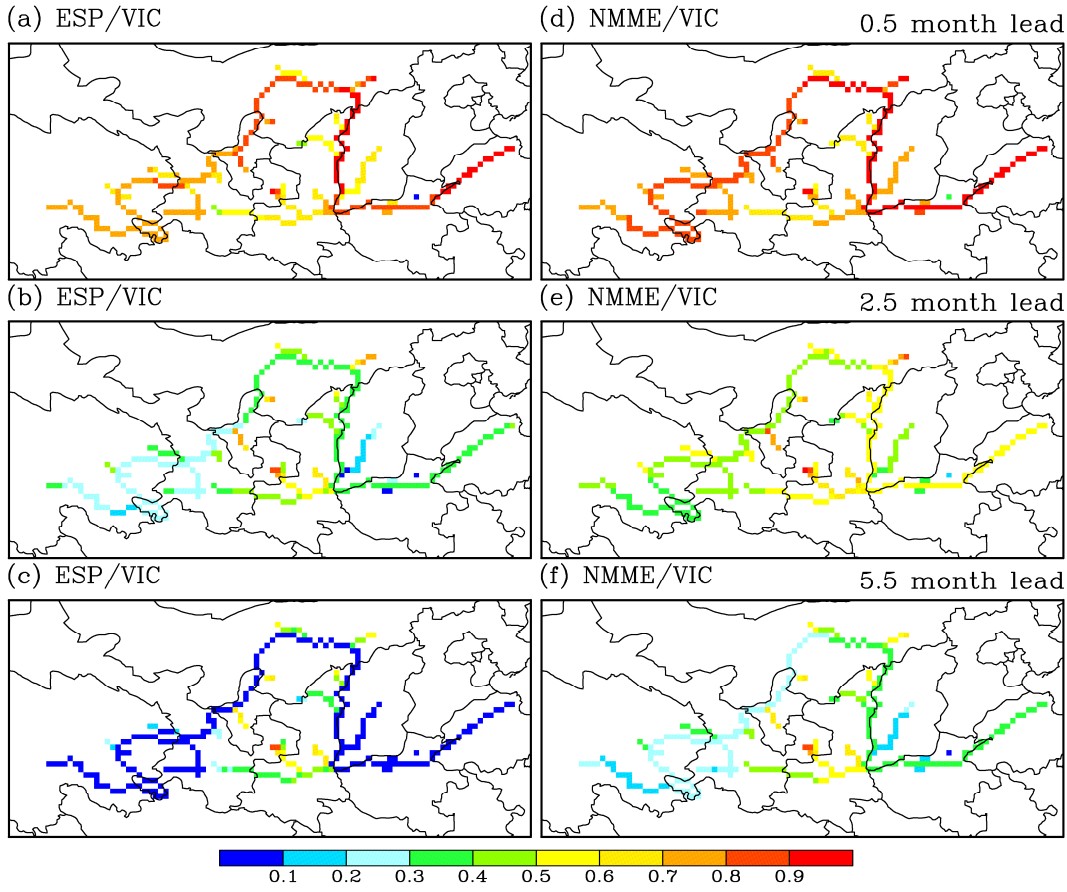

**Figure 6.** The same as Figure 5, but for streamflow along the mainstream and major tributaries of the Yellow River.





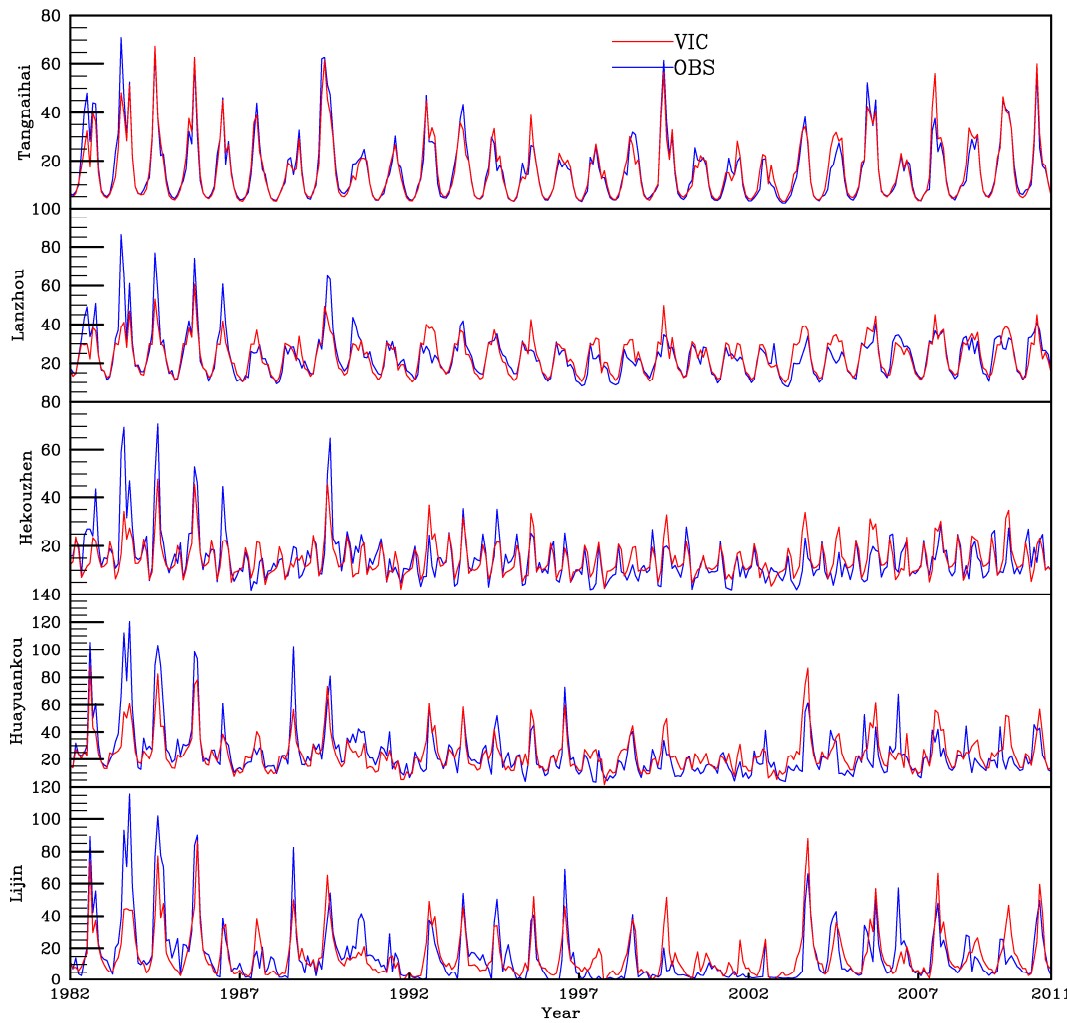

**Figure 7.** Observation (blue) and post-processed VIC simulation (red) for the monthly streamflow ($10^8$ m$^3$/s) at five hydrological gauges located from upper to lower mainstream of the Yellow River. During the post-processing procedure, the simulated streamflow without human interventions is linearly regressed against the observed streamflow for each target month.





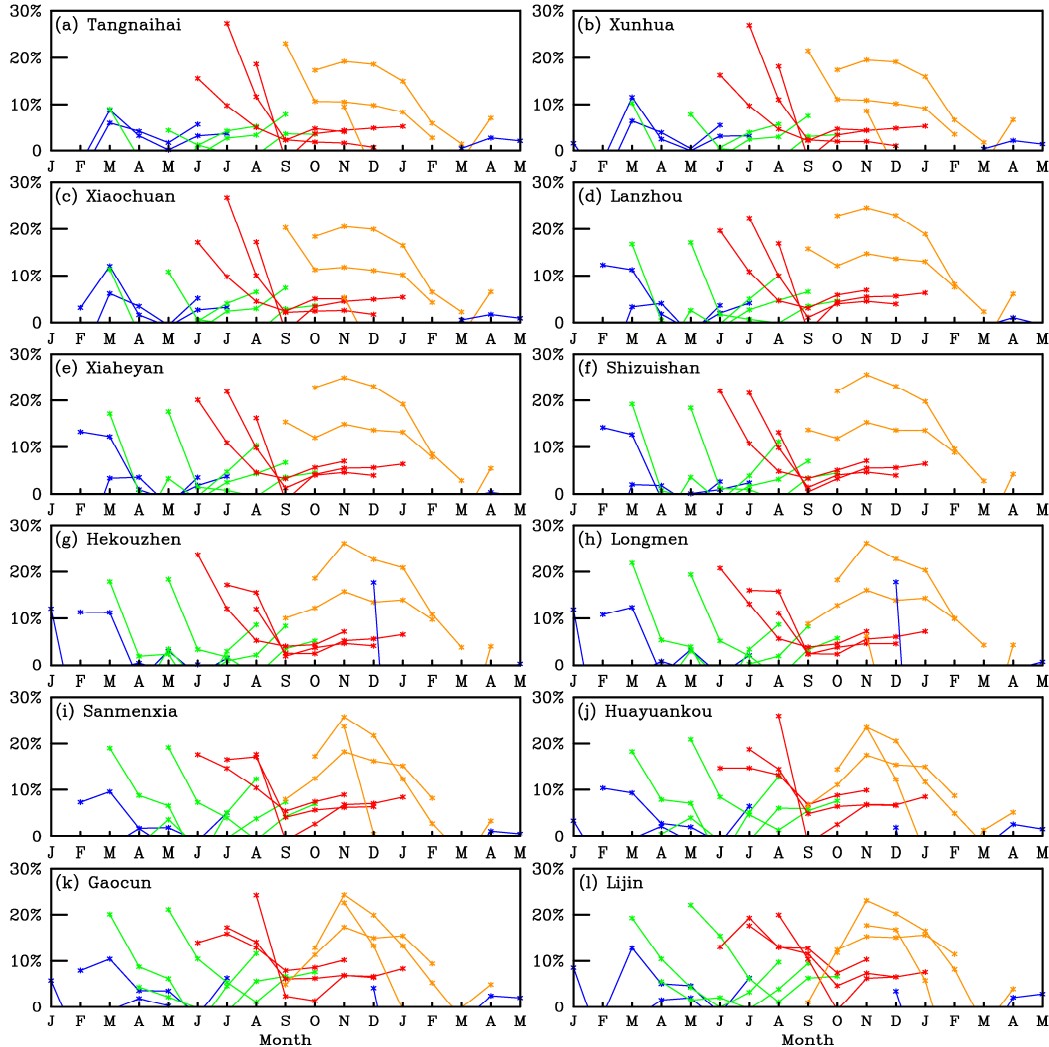

**Figure 8.** The Root Mean Squared Error Skill Score (SS$_{RMSE}$) for streamflow as a function of start month and lead time at twelve hydrological gauges. The SS$_{RMSE}$ is defined as 1-RMSE$_{NMME}$/RMSE$_{ESP}$, where RMSE$_{NMME}$ and RMSE$_{ESP}$ are the RMSEs for the streamflow forecasts from NMME/VIC and ESP/VIC verified against the offline simulated streamflow.





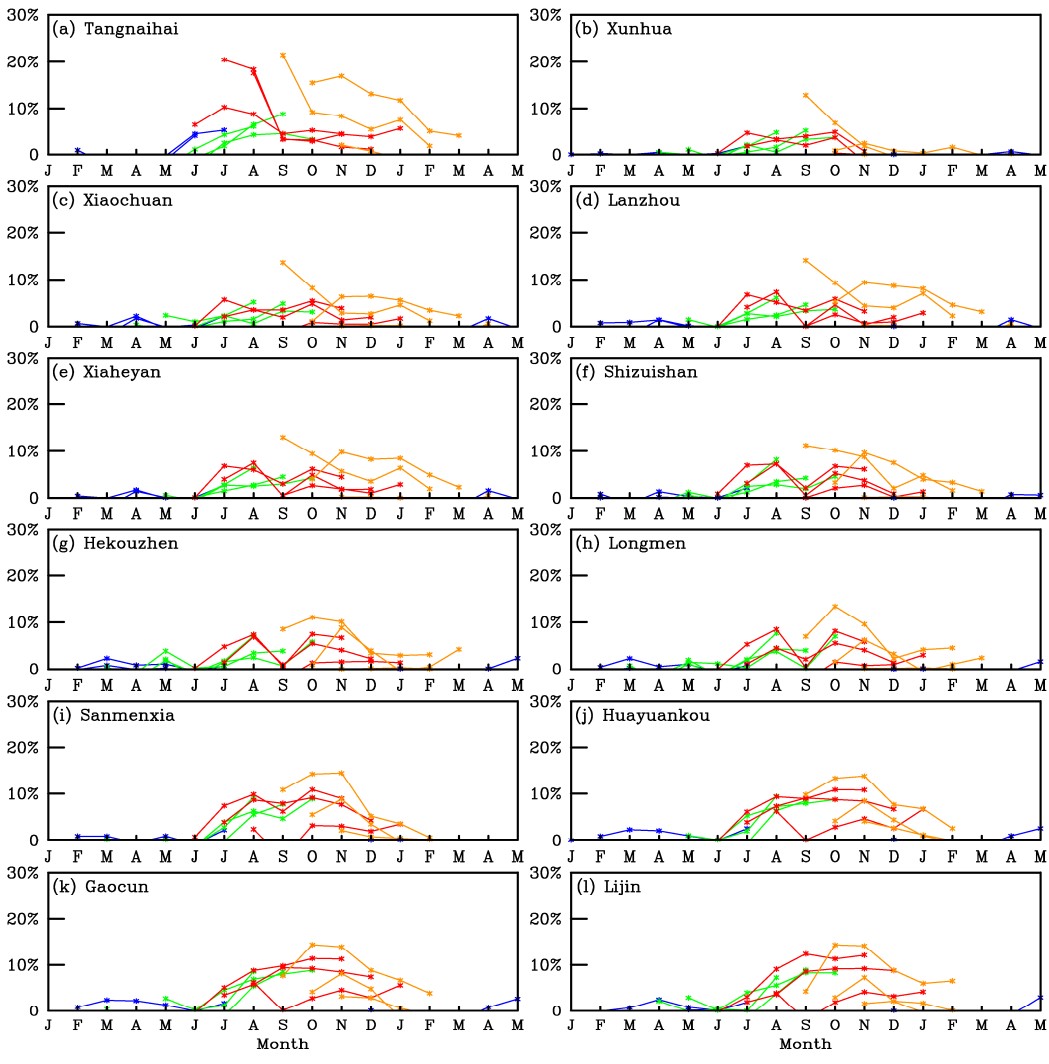

**Figure 9.** The same as Figure 8, but the RMSEs are calculated against observed streamflow.