# Peer review of "An experimental seasonal hydrological forecasting system over the Yellow River basin-Part II: The added value from climate forecast models"

_Hydrology and Earth System Sciences, 2016_

## Referee Comment (RC1) · Anonymous Referee #1 · 2 Apr 2016

I find this study very well carried out and the paper very well written. Following the Part I of the study, the author investigated how much extra forecast skill the NMME ensembles can provide relative to the baseline statistical forecast (ESP) which relies on the initial hydrologic conditions only (no information from dynamic forecast). To my knowledge, NMME has not been looked at over the study area here, the Yellow River basin, and I think the study presented here offered a lot of new insights about NMME and seasonal scale hydrologic forecast in general. So I think the work here is more than enough significant for being considered published at HESS.

The analysis in the paper is focused on the two main drivers for surface hydrology, precipitation and air temperature, as well as two key hydrologic variables, soil moisture

and river streamflow. A land surface model (VIC) and a river routing model were used to derive the surface hdyrologic fluxes/states. The author also applied a number of important techniques like downscaling, bias correction, and post-processing in an effort to maximize the accuracy and skill of the final hydrologic forecasts. There is a solid amount of careful experiments and analyses. Besides the scientific quality, the author has also done a good literature review and the presentation is also well organized.

My main concern is about the technical details of the analysis. The main skill metric used is the Anomaly Correlation, defined in Equation (1) on page 6, as the correlation calculated over both time and space. I think the author needs to offer some reasoning to back up such a definition. Normally, the skill can be defined as the correlation between forecasts and observations in time only. Why to lump all locations together calculating the correlation? Why not calculate the correlation over different locations first and then average them up? I guess that the short length of the data records (29 years) might be a factor which makes the correlation calculations less robust. The current definition lumps all locations together and it is hard to distinguish between NMME's ability to resolve the dynamics in time and space. Because of that, I can't quite interpret some of the discussions later, for example, about the significance of low correlation in lines 5-8 on page 7. If we calculate the correlation over 38309 samples, then the correlation includes both those in time and space ... and in which part shall we measure the forecast skill?

Also, a very minor point – can you show an example of the hydrological post-processing? For example, to the time series of the raw, post-processed, and observed streamflow at one gauge? Did you train the regressions using data over the same period of 1982-2010 or a different period?

Overall, I think the paper can be published in HESS after minor revisions.

---

## Referee Comment (RC2) · Anonymous Referee #2 · 2 Apr 2016

The second of the two papers concerning the establishment of the seasonal ensemble hydrological prediction system in the Yellow River basin, this paper describes the investigation of the added value from implementing the ensemble of climate models into the considered framework.

Two main forecast ensembles are compared: the ESP/VIC approach produces streamflow forecasts based on the ensemble of 28 meteorological conditions from the period of 1982-2010 and an ensemble of 8 North American Multimodel Ensemble models with a total of 99 members (referred to as NMME/VIC). The forecasts of soil moisture and naturalized streamflow are compared using two metrics – Anomaly Correlation and RMSE Skill score. The AC plots show that the NMME/VIC approach may enhance the

forecast skill for both streamflow and soil moisture at longer lead times.

To produce a forecast that would be comparable to the observations, the output from both approaches is then post-processed by a linear regression. The regression coefficients are derived by fitting the naturalized multiannual streamflow time-series to the observed time-series. After the post-processing, the NMME/VIC shows a significant reduction in RMSE as compared to the naturalized streamflow.

Considering a hydrological system with high human interventions, would applying a linear regression for streamflow time-series be the best practice in fitting the simulated streamflow to the obeserved? Would water subtractions be a linear or a non-linear process? Is it possible to introduce a seasonally-dependent water subtraction submodel in the VIC model based on e.g. municipal subtraction statistics and would the whole framework benefit from that?

The reviewer kindly asks the author to provide further insight in section 5 on the reasons for a significant decrease in forecast RMSE skill verified against the observed streamflow. As far as the reviewer have understood, the VIC model was calibrated against the naturalized streamflow and only fit to the observed streamflow by linear regression, so were the forecasts.

With the minor additions the paper is suitable for publication.

Technical corrections: - page 3 line 19: correction "of the simulated streamflow"

---

## Referee Comment (RC3) · Anonymous Referee #3 · 27 Apr 2016

This is the second of the two-paper series on development of a seasonal hydrological forecast system over the Yellow River basin. This paper focuses on the evaluation of added value of NMME climate forecast over the study region. It also briefly examined the impact of the streamflow postprocessing on forecast skills. Overall, the study is simple and straightforward. The same methods have been used in exact way before over other regions by other researchers. So the key contribution of this study is the application over Yellow river basin. The evaluation is also quite simple with only the anomaly correlation and RMSE skill score as the major metrics. The analysis is straightforward, and the conclusions are mostly well supported. There are a number of places where the statement or conclusion is not properly justified, or the interpretation

of the result needs to be thought over. I suggest a minor (towards major) revision, and my specific comments are listed below. 1. Page 2, line 15: A reference is needed here to support this statement. 2. Page 2 line 16: change "flooding forecast" to "flood forecasting". 3. Page 2, line 25: Is NMME qualified to be called "open source"? Its forecasts are made available to the research community, but the system itself is not open source, is it? 4. Page 3, line 28: If Yellow river basin is HEAVILY managed, I wonder if such activities can be simply represented by a linear regression in the post-processing procedure. The probability distribution will be highly distorted as the goal of water resource management over the river is to do flood control and irrigation withdraw. Thus observed flow is much more steady (less variant) with less extremes during both dry and wet conditions. Linear regression is typically used between variables that are normally distributed. Can you commend on this? This the linear regression is not suitable here, it needs to be corrected. 5. Page 4, line 31: why don't you use lapse rate correction here when binlinear interpolate the temperature forecast from models? 6. Page 4 line 32: When you say "all ensemble member", are you referring to all members from one individual model or from the entire NMME ensemble? 7. Page 5 linear 10: Again, I don't think the linear regression model is suitable or good enough to represent the human component of the hydrological system. 8. Table 2: Can you actually show how the two time series of streamflow look like, with a QQ plot or scatter plot? The current illustration is not very convincing. 9. Page 5 line 32: change "measures" to "metrics 10. Equation 1: This equation gives a space-time mixed formula for anomaly correlation. Later in the paper, AC is also calculated for individual location, so it is necessary to mention that equation 1 can be simplified for such purpose. 11. Page 6, line 10: What is the impact of having different ensemble members in ESP and NMME on the skill assessment? 12. Figure 1: A pixelited shaded plot probably looks better and easier to read tan the current one. Can you highlight the correlations that are actually statistically significant? 13. Page 6, line 16: I don't agree with this assessment. Most models do show the highest skill for month 1, maybe except GFDL. So the lead time is still quite important, maybe just as important as seasonality. If you think there is not

dependence on lead time, what could be the cause for that? 14. Page 6, line 27: It would be interesting to know how much of the improvement in forecast skill is due to increase in the ensemble size. 15. Figure 2: The current way of plotting makes the 0.5 month value almost invisible. I suggest a pixelited shaded plot. 16. Page 7, line 23: I don't think you cannot draw conclusion like this from Figure 4 although this is likely very true. Statements like this need to be more careful. 17. Page 7, line 32: "less" than what? 18. Page 8, line 25: "representativeness"?? 19. Page 8, line 34: What non-stationary feature are you referring to here? If there is a trend, can you actually tell if it is caused by water withdraw or climate change? 20. Figure 8: Why not show the negative part of SS? 21. Page 10, line 3: IC is important, but not necessary always dominant. 22. Page 10 line 23-25: This conclusion is counterintuitive. Are you saying that if we were to have a perfect land surface model, the climate forecast in hydrological forecasts at long leads would be less useful? 23. Page 10, line 31-32: This addresses a different type of uncertainty. Use of multiple models help to address uncertainties associated with model, not observations. 24. Page 11, line 3: This depends on what type of downscaling method to be used, a dynamic downscaling scheme might not suffer the same. 25. The last paragraph is an interesting discussion, but some of the statements are not directly based on the results of the current research, might need to be revised somehow.

---

## Author Comment (AC3) · 1 May 2016

I am very grateful to the reviewer for the positive and careful review. The thoughtful comments have helped improve the manuscript. The reviewer's comments are italicized and my responses immediately follow.

1. Page 2, line 15: A reference is neededhere to support this statement.

Response: I will add the references "Pappenberger et al., 2008; Swinbank et al., 2016".

2. Page 2 line 16: change "flooding forecast" to "floodforecasting".

Response:I will revise it as suggested.

[Figure]

3. Page 2, line 25: Is NMME qualified to be called "open source"? Itsforecasts are made available to the research community, but the system itself is notopen source, is it?

Response: I did not say that the NMME models are "open source". I called it "an open source of multimodel seasonal climate hindcast datasets" in the paper. Those hindcast datasets are made available to the public by the IRI personnel through the NMME project.

4. Page 3, line 28: If Yellow river basin is HEAVILY managed, I wonder if such activities can be simply represented by a linear regression in the postprocessing procedure. The probability distribution will be highly distorted as the goal of water resource management over the river is to do flood control and irrigation withdraw. Thus observed flow is much more steady (less variant) with less extremes duringboth dry and wet conditions. Linear regression is typically used between variables thatare normally distributed. Can you commend on this? This the linear regression is notsuitable here, it needs to be corrected.

Response: Thanks for the important comment. Actually the naturalized and observed streamflow datasets do show the characteristics in the upper reaches of the basin as the reviewer's comment: the observed flow is much more steady (less variant) with less extremes during both dry and wet seasons (e.g., Lanzhou station in Figure 7). But this is not the case in the lower reaches, where the observed streamflow is consistently lower than the naturalized streamflow due to heavy human water consumption. To account for the seasonality in the water management, the linear regression is applied for each calendar month, where the water allocations during different years are similar and stable. Therefore, the linear regression method can be used to correct the systematic biases. However, I agree with the reviewer that it has drawbacks for correcting the nonlinear errors, where I mentioned it in the manuscript. With the water allocation and consumption data collected in the future, more sophisticated method should be implemented in the forecasting system. I will revise the manuscript as follows: "For

the upper reaches, the observed flow is much more steady (less variant) with less extremes during both dry and wet seasons; but for the lower reaches, the observed streamflow is consistently lower than the naturalized streamflow due to heavy human water consumption."

And the disadvantage of the linear regression method will be discussed at the end of the paper as follows: "(1) a linear time series post-processing model, although considering the seasonality in the water subtraction by calibrating the parameters against observed streamflow month by month, is not sufficient to simulate and forecast a hydrological system with intensive human interventions because of the nonlinearity and nonstationarity. Either connecting with a seasonally dependent water subtraction submodel based on the subtraction statistics or explicitly representing the human intervention processes in the forecasting system is not only necessary to further reduce the uncertainty in the hydrological models, but also to facilitate the understanding of the hydrological predictability with human dimension".

5. Page 4, line 31: why don't you use lapse rate correction here when binlinear interpolate the temperature forecast from models?

Response: The systematic bias (including the topography induced temperature bias) can be easily corrected by using the quantile-mapping method that is used in this study, i.e., mapping the forecast into the observed climatology with lapse rate correction.

6. Page 4 line 32: When you say "all ensemble member", are you referring to all members from one individual model or from the entire NMME ensemble?

Response: As I mentioned in the manuscript, "for each calendar month and each NMME model..." So it refers to all members from one individual model. The rationale is that different models have different climatology that affects the robustness, so I chose to correct them individually.

7. Page 5 linear 10: Again, I don't think the linear regression model is suitable or good

enough to represent the human component of the hydrological system.

Response: I did not clarify that the linear regression saves the day. The statement actually clarifies that the hydrological post-processing is necessary to bridge the gap between the observed streamflow and a hydrological model calibrated against the naturalized streamflow. As I respond above, the disadvantage of the linear regression model will be discussed. But the linear regression does make the hydrological simulations closer to the observation over the river basins with human interventions.

8. Table 2: Can you actually showhow the two time series of streamflow look like, with a QQ plot or scatter plot? The current illustration is not very convincing.

Response: Thanks for the comment. Actually Fig. 7 shows a few examples of the time series of streamflow from observation and post-processed simulations. And I will add the time series of the simulated streamflow before post-processing and the naturalized streamflow into the same figure for comparison.

9. Page 5 line 32: change "measures" to"metrics

Response: I will revise it as suggested.

10. Equation 1: This equation gives a space-time mixed formula for anomalycorrelation. Later in the paper, AC is also calculated for individual location, so it is necessary to mention that equation 1 can be simplified for such purpose.

Response: As I already mentioned in the manuscript, "If the AC is used for each grid cell within the Yellow River basin (i.e., there is only a summation over time), it is reduced to the Pearson correlation."

11. Page 6, line10: What is the impact of having different ensemble members in ESP and NMME onthe skill assessment?

Response: To my experience, more ensemble members will result in higher reliability, but not necessarily the sharpness. As we know, the ESP forecast refers to a kind of

climatological forecast, so it is already the most reliable forecast. Adding more ensembles to the ESP usually does not improve the skill. Currently, the ESP consists of all historical forcings for the target seasons (excluding the target year) during the validation period (1982-2010), if we expand it with more ensemble members (e.g., those forcings before 1982), it has a risk that the results will be even biased if there is a shift in the climate (e.g., decadal variation). Given that the main focus of the paper is the deterministic forecast skill and the setup of the ESP experiment, I think the impact of the ensemble members for the ESP simulation is limited.

12. Figure 1: A pixelited shaded plot probably looks better and easier to read tan the current one. Can you highlight the correlations that are actually statistically significant?

Response: Thanks for the comment. I will revise as suggested. As I mentioned in the manuscript, an anomaly correlation (AC) of 0.05 (as shown in colors) would be statistically significant give the large samples.

13. Page 6, line 16: I don't agree with this assessment. Most models do show the highest skill for month 1, maybe except GFDL. So the lead time is still quite important, maybe just as important as seasonality. If you think there is not dependence on lead time, what could be the cause for that?

Response: I did not state that the lead time is not important. I used "not necessarily" but not the "NOT" in the speculation. It is related to the strong seasonality, i.e., it is usually more skillful during dry season than wet season.

14. Page 6, line 27: It would be interesting to know how much of the improvement in forecast skill is due to increase in the ensemble size.

Response: The reviewer raises an interesting question. Actually in the past, I did some testing to see whether a subset of the NMME ensemble is more skill than the grand NMME ensemble in terms of the deterministic forecast. But it is very difficult to select the optimal subset ensemble members. For the training period, sometimes a subset

ensemble is more skillful than the grand ensemble; but it usually does not hold for the verification period. For a real-time forecasting, it is very difficult to select a subset of the NMME models (according to the hindcast) that is consistently more skillful than the grand ensemble mean. Perhaps it is partly because of the similarity of the NMME models that we discussed in a paper (Yuan and Wood, 2012). But again, this is very complicated especially for the precipitation, and it is out of the scope of the paper. So I decided not to include it in the paper. If the reviewer has any suggestions, I am very glad to try it in the future study.

15. Figure 2: The current way of plotting makes the 0.5month value almost invisible. I suggest a pixelited shaded plot.

Response: I will revise it as suggested.

16. Page 7, line 23:I don't think you cannot draw conclusion like this from Figure 4 although this is likelyvery true. Statements like this need to be more careful.

Response: Thanks for the comment. In this section, all hindcasts are verified against VIC offline simulations, i.e., the errors in the hydrological forecast model is neglected. To avoid confusion, I will add a note in the revised manuscript: "(but note that this result may be model dependent since the hydrological hindcasts in this section are verified against VIC offline simulations by neglecting the errors in the hydrological model)"

17. Page 7, line 32: "less"than what?

Response: I will revise it as "As the ICs have less control on the runoff forecasts than the meteorological forcings…"

18. Page 8, line 25: "representativeness"??

Response: I will revise as suggested.

19. Page 8, line 34: What non-stationary feature are you referring to here? If there is a trend, can you actually tell if it is caused by water withdraw or climate change?

Response: This refers to the human interventions. I will incorporate the time series of naturalized streamflow and the VIC simulations without post-processing in the revised Figure 7. The revised figure will show that the drying trend in the 1980s and 1990s is both caused by climate change and human interventions because the naturalized streamflow also has a drying trend, but is weaker than the observed streamflow.

20. Figure 8: Why not showthe negative part of SS?

Response: There are some small negative values for SS, but are not significantly different from zero. Therefore, they (as well as those small positive values) are not discussed in the paper for investigating the added value from climate-model-based hydrological forecast.

21. Page 10, line 3: IC is important, but not necessary always dominant.

Response: Thanks for the comment. I will replace it with "strong control".

22. Page 10 line 23-25: This conclusion is counter intuitive. Are you saying that if we were to have a perfect land surface model, the climate forecast in hydrological forecasts at long leads would be less useful?

Response: Most previous studies verify the forecast against hydrological model simulations by neglecting the errors in the hydrological models. My statement is that those studies (NOT perfect hydrological model) might underestimate the usefulness of the climate forecasts at long leads. With a perfect hydrological model, the skill for the climate-model-based hydrological forecasting will increase, and so does for the ESP forecasting, so the added value from (usefulness of) climate model is not necessarily increase.

23. Page 10, line 31-32: This addresses a different type of uncertainty. Use of multiple models help to address uncertainties associated with model, not observations.

Response: What I mean is exactly the same as the reviewer. To avoid confusion, I will revise it as follows: "...forecasting with multiple hydrological models might be useful to

quantify the uncertainty in the hydrological model"

24. Page 11, line 3: This depends on what type of downscaling method to be used, a dynamic downscaling scheme might not suffer the same.

Response: According to my experiences in the dynamical downscaling (Yuan and Liang, 2011; Yuan et al., 2012), neglecting the human component will also affect the performance of dynamical downscaling. This is because most climate models, especially for those used in the seasonal forecasting, do NOT consider the human interventions such as reservoir regulation, irrigation, land use changes and groundwater pumping, and their forecasts may suffer from that.

25. The last paragraph is an interesting discussion, but some of the statements are not directly based on the results of the current research, might need to be revised somehow.

Response: I thank the reviewer for the appreciation. This discussion focus on the representation of human intervention in the hydrological forecasting system, development of the system with multiple hydrological models, prediction of seasonal hydrology within the context of global environmental change, and the interpret of the ensemble hydrological forecast. They are the questions we would like to address in our future study. So I would like to keep them unless the reviewer has specific concerns.

References:

Pappenberger, F., Bartholmes, J., and Thielen, J., et al.: New dimensions in early flood warning across the globe using grand-ensemble weather predictions, Geophys. Res. Lett., 35, L10404, doi:10.1029/2008GL033837, 2008.

Swinbank, R., et al.: The TIGGE project and its achievements, Bull. Am. Meteorol. Soc., 50, 49-67, doi:10.1175/BAMS-D-13-00191.1, 2016.

Yuan, X., and Wood, E. F.: On the clustering of climate models in ensemble seasonal forecasting, Geophysical Research Letters, 39, L18701, doi:10.1029/2012GL052735,

2012

Yuan, X., Liang, X.-Z., and Wood, E. F.: WRF ensemble downscaling seasonal forecasts of China winter precipitation during 1982-2008, Climate Dynamics, 39, 2041-2058, doi:10.1007/s00382-011-1241-8, 2012

Yuan, X., and Liang, X.-Z., 2011: Improving cold season precipitation prediction by the nested CWRF-CFS system, Geophysical Research Letters, 38, L02706, doi:10.1029/2010GL046104.

---

## Author Response (AR1)

[revised manuscript text omitted]

Email: yuanxing@tea.ac.cn
Tel: +86-10-82995385
http://www.escience.cn/people/yuanxing

May 24, 2016

Dr. Alexander Gelfan
Editor
Hydrology and Earth System Sciences

RE: manuscript #hess-2016-102

Dear Dr. Gelfan,

Thank you for your kind decision letter on our manuscript entitled "An experimental seasonal hydrological forecasting system over the Yellow River basin-Part II: The added value from climate forecast models**"** (hess-2016-102). We have carefully considered your and reviewer's comments and incorporated them into the revised manuscript to the extent possible. We hope that you find the revised manuscript and the response to the reviews acceptable to *HESS*.
The detailed responses to the comments are attached.

We appreciate the effort you spent to process the manuscript and look forward to hearing from you soon.

Sincerely yours,

Xing Yuan

**Responses to the comments from Reviewer #1**

*I find this study very well carried out and the paper very well written. Following the Part I of the study, the author investigated how much extra forecast skill the NMME ensembles can provide relative to the baseline statistical forecast (ESP) which relies on the initial hydrologic conditions only (no information from dynamic forecast). To my knowledge, NMME has not been looked at over the study area here, the Yellow River basin, and I think the study presented here offered a lot of new insights about NMME and seasonal scale hydrologic forecast in general. So I think the work here is more than enough significant for being considered published at HESS. The analysis in the paper is focused on the two main drivers for surface hydrology, precipitation and air temperature, as well as two key hydrologic variables, soil moisture and river streamflow. A land surface model (VIC) and a river routing model were used to derive the surface hydrologic fluxes/states. The author also applied a number of important techniques like downscaling, bias correction, and post-processing in an effort to maximize the accuracy and skill of the final hydrologic forecasts. There is a solid amount of careful experiments and analyses. Besides the scientific quality, the author has also done a good literature review and the presentation is also well organized.*

**Response**: I would like to thank the reviewer for the compliment and recognizing the value of our work. The thoughtful comments have helped improve the manuscript. The reviewer's comments are italicized and my responses immediately follow.

*My main concern is about the technical details of the analysis. The main skill metric used is the Anomaly Correlation, defined in Equation (1) on page 6, as the correlation calculated over both time and space. I think the author needs to offer some reasoning to back up such a definition. Normally, the skill can be defined as the correlation between forecasts and observations in time only. Why to lump all locations together calculating the correlation? Why not calculate the correlation over different locations first and then average them up? I guess that the short length of the data records (29 years) might be a factor which makes the correlation calculations less robust. The current definition lumps all locations together and it is hard to distinguish between NMME's ability to resolve the dynamics in time and space. Because of that, I can't quite interpret some of the discussions later, for example, about the significance of low correlation in lines 5-8 on page 7. If we calculate the correlation over 38309 samples, then the correlation includes both those in time and space ... and in which part shall we measure the forecast skill?*

**Response**: Thanks for the comment. The anomaly correlation (AC) that assesses the performance both in space and time is widely used in the evaluations of the hydro-climate forecasts (Becker et al., 2014; Saha et al., 2014; Mo and Lettenmaier, 2014; Ma et al., 2015). The use of the AC facilitates the presentation of the results for different target months over different lead times in a single plot (e.g., Figures 1 and 2). Of course, it can also be reduced to the pearson correlation (time) or the pattern correlation (space). For example, Figure 3 shows the temporal part of the AC for the precipitation forecasts over different locations. As pointed out by the reviewer, the short length of the data records (29 years) might be a factor which makes the temporal correlation less robust. The AC samples the

forecasts both over space and time, and it can be regarded as an integral measure of the performance. To clarify it, I have revised the manuscript as follows:

"The AC is widely used in the hydro-climate forecast evaluations (Becker et al., 2014; Saha et al., 2014; Mo and Lettenmaier, 2014; Ma et al., 2015), and can be regarded as a measure of forecast skill both in space and time. If the AC is used for each grid cell within the Yellow River basin (i.e., there is only a summation over time), it is reduced to the Pearson correlation. And if the AC is used for each year, it is reduced to the pattern correlation." (P6, L8-11 in the tracked version)

*Also, a very minor point – can you show an example of the hydrological postprocessing? For example, to the time series of the raw, post-processed, and observed streamflow at one gauge? Did you train the regressions using data over the same period of 1982-2010 or a different period?*

**Response**: Thanks for the comment. I have now plotted the streamflow time series before and after post-processing in Figure 7. The training is done in a cross validation mode. In addition, I have clarified the post-processing procedure in the revised manuscript as follows:

"In this study, a linear regression is applied to correct the streamflow forecasts at 12 mainstream gauges where the observations are available. For each gauge, the regression coefficients are firstly fitted between observed and offline simulated streamflow for each calendar month to account for the seasonality in the human water usage, then the coefficients are applied to correct the streamflow forecasts for their target months. The coefficients are estimated during 1982-2010 in a cross validation mode (i.e., dropping the target year)." (P5, L12-17)

**References:**

Becker, M., van Den Dool, H., and Zhang, Q: Predictability and forecast skill in NMME, J. Climate, 27, 5891-5906, doi:10.1175/JCLI-D-13-00597.1, 2014.

Ma, F., Yuan, X., and Ye, A.: Seasonal drought predictability and forecast skill over China, J. Geophys. Res. Atmos., 120, 8264–8275, doi:10.1002/2015JD023185, 2015.

Mo, K. C., and Lettenmaier, D. P.: Hydrologic prediction over the conterminous United States using the National Multi-Model Ensemble, J. Hydrometeorol., 15, 1457-1472, doi: 10.1175/JHM-D-13-0197.1, 2014.

Saha, S., et al.: The NCEP climate forecast system version 2, J. Climate, 27, 2185-2208, doi:10.1175/JCLI-D-12-00823.1, 2014.

**Responses to the comments from Reviewer #2**

*The second of the two papers concerning the establishment of the seasonal ensemble hydrological prediction system in the Yellow River basin, this paper describes the investigation of the added value*
5 *from implementing the ensemble of climate models into the considered framework. Two main forecast ensembles are compared: the ESP/VIC approach produces streamflow forecasts based on the ensemble of 28 meteorological conditions from the period of 1982-2010 and an ensemble of 8 North American Multimodel Ensemble models with a total of 99 members (referred to as NMME/VIC). The forecasts of soil moisture and naturalized streamflow are compared using two metrics – Anomaly Correlation and*
10 *RMSE Skill score. The AC plots show that the NMME/VIC approach may enhance the forecast skill for both streamflow and soil moisture at longer lead times. To produce a forecast that would be comparable to the observations, the output from both approaches is then post-processed by a linear regression. The regression coefficients are derived by fitting the naturalized multiannual streamflow time-series to the observed time-series. After the post-processing, the NMME/VIC shows a significant*
15 *reduction in RMSE as compared to the naturalized streamflow.*

**Response**: I would like to thank the reviewer for the compliment and recognizing the value of our work. The thoughtful comments have helped improve the manuscript. The reviewer's comments are italicized and my responses immediately follow.

20 *Considering a hydrological system with high human interventions, would applying a linear regression for streamflow time-series be the best practice in fitting the simulated streamflow to the obeserved? Would water subtractions be a linear or a non-linear process? Is it possible to introduce a seasonally-dependent water subtraction submodel in the VIC model based on e.g. municipal subtraction statistics and would the whole framework benefit from that?*

25 **Response**: Thanks for the important comment. I agree with reviewer that a linear regression is not sufficient to account for the nonlinearity and nonstationarity in the hydrological system with intensive human interventions. Incorporating a water subtraction submodel is a good suggestion for future work. Currently, the post-processing with linear regression is applied for each calendar month, so the seasonality in water subtraction can be addressed to some extent. I have incorporated the reviewer's
30 comment into the discussion section as follows:

"(1) a linear time series post-processing model, although considering the seasonality in the water subtraction by calibrating the parameters against observed streamflow month by month, is not sufficient to simulate and forecast a hydrological system with intensive human interventions because of the nonlinearity and nonstationarity. Either connecting with a seasonally-dependent water subtraction sub-
35 model based on the subtraction statistics or explicitly representing the human intervention processes in the forecasting system is not only necessary to further reduce the uncertainty in the hydrological models, but also to facilitate the understanding of the hydrological predictability with human dimension;" (P11, L8-14 in the tracked version)

*The reviewer kindly asks the author to provide further insight in section 5 on the reasons for a significant decrease in forecast RMSE skill verified against the observed streamflow. As far as the reviewer have understood, the VIC model was calibrated against the naturalized streamflow and only fit to the observed streamflow by linear regression, so were the forecasts.*

5 **Response**: Thanks for the suggestion. I have revised the manuscript as follows:
"The decrease in the RMSE skill score is consistent with previous finding over the USA (Yuan et al., 2013), which is because of the increase in the uncertainty of hydrological models. Given that the VIC model used in this study has no parameterization in the human water consumption, a linear regression in the post-processing procedure may reduce the systematic bias with the consideration of seasonality, but

10 it does not necessarily correct the errors in the variability. Connecting the VIC model with water subtraction model with different complexities (e.g., from statistical to process-based models) will reduce the uncertainty in the hydrological model, and thus amplify the add value from climate forecast models." (P9, L16-22)

15 *With the minor additions the paper is suitable for publication.*

*Technical corrections: - page 3 line 19: correction "of the simulated streamflow"*
**Response**: Revised as suggested. (p3, L19)

**Responses to the comments from Reviewer #3**

I am very grateful to the reviewer for the positive and careful review. The thoughtful comments have helped improve the manuscript. The reviewer's comments are italicized and my responses immediately follow.

1.  *Page 2, line 15: A reference is neededhere to support this statement.*

**Response**: I have added the references "Pappenberger et al., 2008; Swinbank et al., 2016". (P2, L13,15 in the tracked version)

2.  *Page 2 line 16: change "flooding forecast" to "floodforecasting".*

**Response**: Revised as suggested. (P2, L16)

3.  *Page 2, line 25: Is NMME qualified to be called "open source"? Itsforecasts are made available to the research community, but the system itself is notopen source, is it?*

**Response**: I did not say that the NMME models are "open source". I called it "an open source of multimodel seasonal climate hindcast datasets" in the paper. Those hindcast datasets are made available to the public by the IRI personnel through the NMME project.

4.  *Page 3, line 28: If Yellow river basin is HEAVILY managed, I wonder if such activities can be simply represented by a linear regression in the postprocessing procedure. The probability distribution will be highly distorted as the goal of water resource management over the river is to do flood control and irrigation withdraw. Thus observed flow is much more steady (less variant) with less extremes duringboth dry and wet conditions. Linear regression is typically used between variables thatare normally distributed. Can you commend on this? This the linear regression is notsuitable here, it needs to be corrected.*

**Response**: Thanks for the important comment. Actually the naturalized and observed streamflow datasets do show the characteristics in the upper reaches of the basin as the reviewer's comment: the observed flow is much more steady (less variant) with less extremes during both dry and wet seasons (e.g., Lanzhou station in the revised Figure 7). But this is not the case in the lower reaches, where the observed streamflow is consistently lower than the naturalized streamflow due to heavy human water consumption. To account for the seasonality in the water management, the linear regression is applied for each calendar month, where the water allocations during different years are similar and stable. Therefore, the linear regression method can be used to correct the systematic biases. However, I agree with the reviewer that it has drawbacks for correcting the nonlinear errors, where I mentioned it in the manuscript. With the water allocation and consumption data collected in the future, more sophisticated method should be implemented in the forecasting system. I have revised the manuscript as follows:

"For the upper reaches, the observed flow is much more steady (less variant) with less extremes during both dry and wet seasons; but for the lower reaches, the observed streamflow is consistently lower than the naturalized streamflow due to heavy human water consumption." (P3, L28-30)

5 And the disadvantage of the linear regression method has been discussed at the end of the paper as follows:

"(1) a linear time series post-processing model, although considering the seasonality in the water subtraction by calibrating the parameters against observed streamflow month by month, is not sufficient to simulate and forecast a hydrological system with intensive human interventions because of the
10 nonlinearity and nonstationarity. Either connecting with a seasonally dependent water subtraction sub-model based on the subtraction statistics or explicitly representing the human intervention processes in the forecasting system is not only necessary to further reduce the uncertainty in the hydrological models, but also to facilitate the understanding of the hydrological predictability with human dimension". (P11, L8-14)

*5. Page 4, line 31: why don't you use lapse rate correction here when binlinear interpolate the temperature forecast from models?*

**Response**: The systematic bias (including the topography induced temperature bias) can be easily corrected by using the quantile-mapping method that is used in this study, i.e., mapping the forecast into
20 the observed climatology with lapse rate correction.

*6. Page 4 line 32: When you say "all ensemble member", are you referring to all members from one individual model or from the entire NMME ensemble?*

**Response**: As I mentioned in the manuscript, "for each calendar month and each NMME model…" (P4,
25 L36 in the tracked version). So it refers to all members from one individual model. The rationale is that different models have different climatology that affects the robustness, so I chose to correct them individually.

*7. Page 5 linear 10: Again, I don't think the linear regression model is suitable or good enough to*
30 *represent the human component of the hydrological system.*

**Response**: I did not clarify that the linear regression saves the day. The statement actually clarifies that the hydrological post-processing is necessary to bridge the gap between the observed streamflow and a hydrological model calibrated against the naturalized streamflow. As I respond above, the disadvantage of the linear regression model will be discussed. But the linear regression does make the hydrological
35 simulations closer to the observation over the river basins with human interventions.

8. *Table 2: Can you actually showhow the two time series of streamflow look like, with a QQ plot or scatter plot? The current illustration is not very convincing.*

**Response**: Thanks for the comment. Actually Figure 7 shows a few examples of the time series of streamflow from observation and post-processed simulations. And I have added the time series of the simulated streamflow before post-processing and the naturalized streamflow into the same figure for comparison. Please see the revised Figure 7 in the revised manuscript.

9. *Page 5 line 32: change "measures" to"metrics*

**Response**: Revised as suggested. (P6, L3)

10. *Equation 1: This equation gives a space-time mixed formula for anomalycorrelation. Later in the paper, AC is also calculated for individual location, so it is necessary to mention that equation 1 can be simplified for such purpose.*

**Response**: I agree with the reviewer, please see P6L9-10: "If the AC is used for each grid cell within the Yellow River basin (i.e., there is only a summation over time), it is reduced to the Pearson correlation."

11. *Page 6, line10: What is the impact of having different ensemble members in ESP and NMME onthe skill assessment?*

**Response**: To my experience, more ensemble members will result in higher reliability, but not necessarily the sharpness. As we know, the ESP forecast refers to a kind of climatological forecast, so it is already the most reliable forecast. Adding more ensembles to the ESP usually does not improve the skill. Currently, the ESP consists of all historical forcings for the target seasons (excluding the target year) during the validation period (1982-2010), if we expand it with more ensemble members (e.g., those forcings before 1982), it has a risk that the results might be biased if there is a shift in the climate (e.g., decadal variation). Given that the main focus of the paper is the deterministic forecast skill and the setup of the ESP experiment, I think the impact of the ensemble members for the ESP simulation is limited.

12. *Figure 1: A pixelited shaded plot probably looks better and easier to read tan the current one. Can you highlight the correlations that are actually statistically significant?*

**Response**: Thanks for the comment. I have revised it as suggested (please see Figure 1 in the revised manuscript). As I mentioned in the manuscript, an anomaly correlation (AC) of 0.05 (as shown in colors) would be statistically significant, given the large samples.

*13. Page 6, line 16: I don't agree with this assessment. Most models do show the highest skill for month 1, maybe except GFDL. So the lead time is still quite important, maybe just as important as seasonality. If you think there is not dependence on lead time, what could be the cause for that?*

**Response**: I did not state that the lead time is not important. I used "not necessarily" but not the "NOT" in the speculation. It is related to the strong seasonality, i.e., it is usually more skillful during dry season than wet season.

*14. Page 6, line 27: It would be interesting to know how much of the improvement in forecast skill is due to increase in the ensemble size.*

**Response**: The reviewer raises an interesting question. Actually in the past, I did some testing to see whether a subset of the NMME ensemble is more skill than the grand NMME ensemble in terms of the deterministic forecast. But it is very difficult to select the optimal subset ensemble members. For the training period, sometimes a subset ensemble is more skillful than the grand ensemble; but it usually does not hold for the verification period. For a real-time forecasting, it is very difficult to select a subset of the NMME models (according to the hindcast) that is consistently more skillful than the grand ensemble mean. Perhaps it is partly because of the similarity of the NMME models that we discussed in a paper (Yuan and Wood, 2012). But again, this is very complicated especially for the precipitation, and it is out of the scope of the paper. So I decided not to include it in the paper. If the reviewer has any suggestions, I am very glad to try it in the future study.

*15. Figure 2: The current way of plotting makes the 0.5month value almost invisible. I suggest a pixelited shaded plot.*

**Response**: I have revised it as suggested. Please see Figure 2 in the tracked version of the revised manuscript.

*16. Page 7, line 23:I don't think you cannot draw conclusion like this from Figure 4 although this is likelyvery true. Statements like this need to be more careful.*

**Response**: Thanks for the comment. In this section, all hindcasts are verified against VIC offline simulations, i.e., the errors in the hydrological forecast model is neglected. To avoid confusion, I have added a note in the revised manuscript:

"(but note that this result may be model dependent since the hydrological hindcasts in this section are verified against VIC offline simulations by neglecting the errors in the hydrological model)" (P7, L29-31)

*17. Page 7, line 32: "less"than what?*

**Response**: I have revised it as "As the ICs have less control on the runoff forecasts than the meteorological forcings…" (P8, L6-7)

*18. Page 8, line 25: "representativeness"??*

**Response**: Revised as suggested. (P8, L33)

*19. Page 8, line 34: What non-stationary feature are you referring to here? If there is a trend, can you actually tell if it is caused by water withdraw or climate change?*

**Response**: This refers to the human interventions. I have incorporated the time series of naturalized streamflow and the VIC simulations without post-processing in the revised Figure 7. The revised figure shows that the drying trend in the 1980s and 1990s is both caused by climate change and human interventions because the naturalized streamflow also has a drying trend, although it is weaker than the observed streamflow.

*20. Figure 8: Why not showthe negative part of SS?*

**Response**: There are some small negative values for SS, but are not significantly different from zero. Therefore, they (as well as those small positive values) are not discussed in the paper for investigating the added value from climate-model-based hydrological forecast.

*21. Page 10, line 3: IC is important, but not necessary always dominant.*

**Response**: Thanks for the comment. I have replaced it with "strong control". (P10, L17)

*22. Page 10 line 23-25: This conclusion is counter intuitive. Are you saying that if we were to have a perfect land surface model, the climate forecast in hydrological forecasts at long leads would be less useful?*

**Response**: Most previous studies verify the forecast against hydrological model simulations by neglecting the errors in the hydrological models. My statement is that those studies (NOT perfect hydrological model) might underestimate the usefulness of the climate forecasts at long leads. With a perfect hydrological model, the skill for the climate-model-based hydrological forecasting will increase, and so does for the ESP forecasting, so the added value from climate model (against) is not necessarily increase.

*23. Page 10, line 31-32: This addresses a different type of uncertainty. Use of multiple models help to address uncertainties associated with model, not observations.*

**Response**: What I mean is exactly the same as the reviewer. To avoid confusion, I have revised it as follows: "…forecasting with multiple hydrological models might be useful to quantify the uncertainty in the hydrological model" (P11, L15-16)

5   *24. Page 11, line 3: This depends on what type of downscaling method to be used, a dynamic downscaling scheme might not suffer the same.*

**Response**: According to my experiences in the dynamical downscaling (Yuan and Liang, 2011; Yuan et al., 2012), neglecting the human component will also affect the performance of dynamical downscaling. This is because most climate models, especially for those used in the seasonal forecasting, do NOT
10 consider the human interventions such as reservoir regulation, irrigation, land use changes and groundwater pumping, and their forecasts may suffer from that.

  *25. The last paragraph is an interesting discussion, but some of the statements are not directly based on the results of the current research, might need to be revised somehow.*

15 **Response**: I thank the reviewer for the appreciation. This discussion focus on 1) the representation of human intervention in the hydrological forecasting system, 2) development of the system with multiple hydrological models, 3) prediction of seasonal hydrology within the context of global environmental change, and 4) the interpret of the ensemble hydrological forecast. They are the questions we would like to address in our future study. So I would like to keep them unless the reviewer has specific concerns.

---

## Editor Decision (ED1)

The authors have taken into account the criticisms and suggestions of both referees that resulted in improvement of the paper. I recommend the revised paper for publication as is.

| Principal Criteria | Excellent (1) | Good (2) | Fair (3) | Poor (4) |
|---|---|---|---|---|
| **Scientific Significance:** Does the manuscript represent a substantial contribution to scientific progress within the scope of *Hydrology and Earth System Sciences* (substantial new concepts, ideas, methods, or data)? | | + | | |
| **Scientific Quality:** Are the scientific approach and applied methods valid? Are the results discussed in an appropriate and balanced way (consideration of related work, including appropriate references)? | | + | | |
| **Presentation Quality:** Are the scientific results and conclusions presented in a clear, concise, and well-structured way (number and quality of figures/tables, appropriate use of English language)? | | + | | |

Alexander Gelfan

Handling Editor